METHODS

# High-throughput sequencing of EcoWI restriction fragments maps the genome-wide landscape of phosphorothioate modification at base resolution

**Weiwei Yang, Alexey Fomenkov, Dan Heiter, Shuang-yong Xu\*, Laurence Ettwiller📧\***

New England Biolabs Inc., Ipswich, Massachusetts, United States of America

\* xus@neb.com (SYX); ettwiller@neb.com (LE)

**Data Availability Statement:** All raw and processed sequencing data generated in this study have been submitted to the NCBI Sequence Read

## Abstract

Phosphorothioation (PT), in which a non-bridging oxygen is replaced by a sulfur, is one of the rare modifications discovered in bacteria and archaea that occurs on the sugar-phosphate backbone as opposed to the nucleobase moiety of DNA. While PT modification is widespread in the prokaryotic kingdom, how PT modifications are distributed in the genomes and their exact roles in the cell remain to be defined. In this study, we developed a simple and convenient technique called EcoWI-seq based on a modification-dependent restriction endonuclease to identify genomic positions of PT modifications. EcoWI-seq shows similar performance than other PT modification detection techniques and additionally, is easily scalable while requiring little starting material. As a proof of principle, we applied EcoWI-seq to map the PT modifications at base resolution in the genomes of both the *Salmonella enterica* cerro 87 and *E. coli* expressing the *dnd+* gene cluster. Specifically, we address whether the partial establishment of modified PT positions is a stochastic or deterministic process. EcoWI-seq reveals a systematic usage of the same subset of target sites in clones for which the PT modification has been independently established.

## Author summary

Large number of bacteria have modified their DNA mainly as part of a strategy to resist virus infection. Most of the modifications are chemical variations on the canonical bases A, T, C or G with phosphorothioate (PT) being a rare exception of a modification that happens on the backbone of the DNA. Interestingly, this PT modification was first chemically synthesized for specific biotechnological processes before scientists discovered that bacteria and archaea naturally perform this modification using their enzymes. The exact roles of phosphorothioation in bacteria and archaea is still under investigation. To enable further investigation of PT modifications, we designed EcoWI-seq, a method to identify the exact positions of these modifications in bacterial genomes. Notably, we applied the EcoWI-seq to several strains of *E. coli* for which PT modification has been induced by cloning into these strains, the necessary genes for making such modification. We found

Archive (SRA; https://www.ncbi.nlm.nih.gov/sra) under accession number PRJNA837277.

**Funding:** This work was supported by New England Biolabs (WY, AF, DH, SX, LE). WY, AF, DH, SX, LE received a salary from the funder. The funder had no role in study design, data collection and analysis, decision to publish, or preparation of the manuscript.

**Competing interests:** I have read the journal's policy and the authors of this manuscript have the following competing interests: WY, AF, DH, SX, and LE are employees of New England Biolabs, Inc, a manufacturer of restriction enzymes and molecular biology reagents.

that these strains, despite being independently made, followed a precise pattern of PT modification with always the same sites being modified. This result indicates a deterministic process for the establishment of PT modification.

## Introduction

Phosphorothioate (abbreviation: PT or ps) backbone modification, where a non-bridging oxygen is replaced by a sulfur atom [1], has been used in biotechnological applications for decades mainly to confer to phosphorothioated oligonucleotides resistance to nuclease digestion and longevity in therapeutic applications [2]. This DNA modification has been recently found naturally occurring in a diversity of bacteria and archaea [3]. The ability to make phosphorothioate DNA (Dnd+) *in vivo* is conferred by the gene products of the *dndABCDE* cluster, typically with *dndBCDE* located on a single operon and *dndA* either located adjacent to *dndBCDE* or scattered elsewhere [4]. Furthermore DndA protein can be functionally substituted by NifS-like cysteine desulfurase IscS in some bacterial strains including *E.coli* K12 [5]. Possible biological functions of PT modifications on DNA are considered as: 1) epigenetic marker to regulate gene expression [6–8]; 2) providing a resistance mechanism to restriction systems or exonuclease degradation [9,10]; 3) an antioxidant to resist intracellular oxidative stress [11–13]; 4) a marker for self vs non-self (here PT-modified DNA as "foreign" and subject to SBD-HNH restriction) [14,15]. So far, a limited number of sequence contexts such as GpsAAC and its reverse complement GpsTTC (GpsAAC/GpsTTC), GpsGCC/GpsGCC, GpsATC/GpsATC, or CpsCA/TGG [3,4] have been found to be PT modified. Furthermore, only a small fraction of these sites is modified, in either one or both strands. For example, less than 20% of available GAAC/GTTC sites are PT-modified in the Dnd+ genomes of *E. coli* B7A (*Eco*B7A) and *Salmonella enterica* serovar Cerro 87 (*Sen*C87) [6,16,17]. How the sites are selected for full, hemi- or no modification is currently unknown.

The past half decade has witnessed the accelerated investigations into characterizing the unique features of phosphorothioation and understanding of its biological functions. These advances are largely benefiting from the increasing capability in detection and profiling of DNA PT modifications in genomic DNA. Current approaches can be grouped into three categories: 1) liquid chromatography coupled with mass spectrometry (LC/MS), which can analyze the PT modification level from enzymatically digested dinucleotides NpsN. While quantitative, this LC/MS-based method does not provide a base resolution distribution of PT positions in the genome [1]; 2) single molecule real-time (SMRT) sequencing enables genomic mapping of phosphorothioation by virtue of variations in the DNA polymerase kinetics, however, with relatively low detection signals and difficulties in discriminating PT specific signals from noise or other base modifications [16]; 3) short reads sequencing of iodine induced cleavage at phosphorothioate sites. These methods such as ICDS (deep sequencing of iodine-induced cleavage at PT), PT-IC-seq and Nick-seq are effective in detecting dsDNA PT modifications (Nick-seq can detect single strand modification) but they require labor intensive preparation procedures [16–18]. A method using optical visualization of PT modifications in single DNA molecules has been developed, however, this method is not high-throughput and doesn't provide base resolution detection [19]. More recently, a PT-site detection method using nanopore sequencing was reported for plasmid and bacterial genomic DNA (gDNA) [20]. A convenient and accurate method for detecting PT modifications is still greatly needed to determine and unravel the unprecedented phosphorothioate epigenomes.

Towards this end, we developed a high throughput sequencing method called EcoWI-seq based on the specific cleavage pattern of EcoWI. EcoWI is a modification-dependent restriction endonucleases (MDREs) that targets the phosphorothioate DNA sequence GpsAAC/GpsTTC and cleaves 3' downstream of its recognition sequence at N7/N6 [21]. MDREs are restriction enzymes that restrict DNA when the bases on DNA substrates are modified as opposed to the classical Type I, II, and III restriction systems for which base modifications diminish or totally abolish their restriction activities [22–24]. EcoWI enzyme belongs to a sub-group of MDREs [25] that contains a sulfur binding domain (SBD) followed by a HNH endonuclease domain to cleave DNA substrates with PT modification.

Our goal is to use EcoWI to digest Dnd+ modified gDNA and sequence the digested fragments by next-generation sequencing (NGS) using Illumina DNA sequencing platform. The reads generated by the enzymatic digestions were mapped to the reference genome to achieve a profile of PT modified sites in the *Salmonella enterica Sen*C87 gDNA. EcoWI-seq shows substantial agreement (98% of common PT modified sites) with the established SMRT sequencing. The advantages of EcoWI-seq over other PT-site mapping methods are low DNA input for library construction, ease of EcoWI digestion similar to other typical restriction digests, and combination with the *de novo* assembly of unknown bacteria strains. We applied EcoWI-seq to address the question whether the establishment of modified PT positions is a stochastic or deterministic process.

## Results

### 1. Development of EcoWI-seq as a one-step enzymatic approach for genome-wide base resolution PT modification mapping

In a previous work [21], we characterized the phosphorothioate-dependent restriction endonuclease (PTDR) EcoWI which cleaves at a fixed distance from its recognition sequence GpsAAC/GpsTTC. The enzyme shows specific and consistent activity on PT-modified GpsAAC/GpsTTC motifs in synthesized oligonucleotides and plasmids. Based on these findings, we developed a high throughput sequencing method named EcoWI-seq to identify genome-wide instances of PT modification at GAAC/GTTC motif. The general concept of the method is illustrated in Fig 1. Briefly, the one-step EcoWI treatment introduces a nicking site seven bases downstream of the PT GpsTTC motif and a nicking site 6 bases upstream of a PT GpsAAC motif thus creating a double-strand break with a single nucleotide 3' end overhang. This digestion results in the pile-ups of fragments ending/starting specifically at the enzyme-induced double-strand break. It has been previously reported that digestion with EcoWI of PT modified organisms results in a broad range of fragment sizes including very large fragments [21]. Thus, to ensure uniform coverage of the genome and a quantifiable representation of all the cut sites irrespective of the cutting frequencies, we subsequently subjected the digested DNA to random fragmentation by sonication. After standard Illumina library preparation and sequencing, the digested sites are distinguished from random fragmentation patterns using a newly developed analytical procedure.

### 2. Identification of PT modified positions by EcoWI-seq at single base resolution in the Salmonella enterica serovar Cerro 87 (SenC87) genome

We applied EcoWI-seq to analyze PT modifications in the wild type (WT) *Salmonella enterica* serovar Cerro 87 (*Sen*C87) genome (Dnd+). This genome has been previously demonstrated to have PT modification at GAAC/GTTC motif (between G and A or T) [3] and a number of studies have characterized the PT modification positions in the genome using SMRT

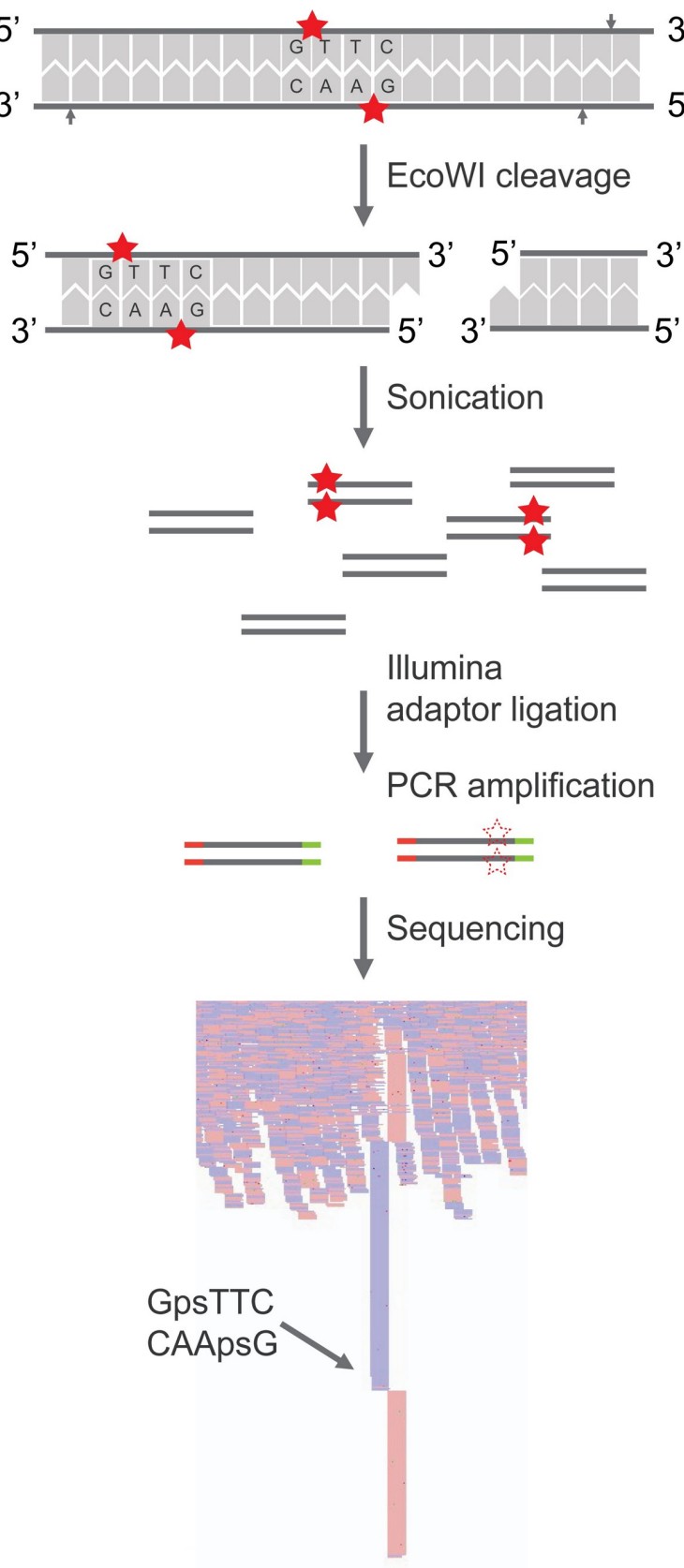

**Fig 1. Schematic illustration of EcoWI-seq.** The figure showed an example of a 5' GpsTTC site (complement GpsAAC). Solid stars represented the PT modifications. EcoWI digestion introduced a double-strand break resulting from two opposite cut sites (indicated by arrows) located at N7/N6 downstream of the PT modified site. Additional cut (s) may occur upstream of a PT GpsTTC motif and/or downstream of a PT GpsAAC motif resulting in either an additional nick or cut respectively [21]. In the former case, the nick will be repaired during the end-prep step during library preparation and will not interfere with the double-strand cleavage pattern (see discussion). The digested gDNA is further sonicated into fragments to obtain appropriately-sized fragments before adaptor ligation. The resulting library is then amplified and sequenced using Illumina platform. Dashed stars showed amplification of fragments generated by EcoWI cuts. The EcoWI cleavage sites resulted in pile-ups of reads start/end that can be differentiated from random breaks generated from sonication.

sequencing and/or iodine-induced sequencing method [16,17]. Additionally, a control experiment was performed on the same starting material using the EcoWI-seq protocol with the omission of the EcoWI enzyme (no-enzyme control). To evaluate the enzyme specificity for PT modification, we also performed EcoWI-seq experiment paired with a no-enzyme control on a *dnd* mutant *Sen*C87 strain for which the *dndE* gene has been deleted (Δ*dndE*) and thus bears no PT modification in its genome (Dnd-). We performed EcoWI-seq on 1 μg of genomic DNA extracted from a 50 ml culture grown on LB medium to stationary phase. Sequencing of all libraries was performed on Illumina instruments and resulted in ~ 10 million paired-end reads per library. Reads were mapped to the *Sen*C87 reference genome and trimmed to leave the most 5' end mapped base.

We showed that nucleotide bias upstream and downstream of positions with high read pile-ups (>50 5' ends) match the GAAC and GTTC motif in the genome of the Dnd+ WT strain, which is consistent with the reported enzyme specificity (Figs 2A and S1A). In contrast, no read pile-ups were observed in the no enzyme control, indicating that the pile-ups of 5' end of reads at specific positions are resulting from the enzyme activity (S1A Fig). We confirmed that read pile-ups remained at low base levels in the genome of the Δ*dndE* mutant strain (S1A Fig), indicating that PT modifications are necessary for the enzyme activity. These results together demonstrate that EcoWI selectively recognizes and cuts PT modified GpsAAC/GpsTTC sites in the genome. These results also indicate that the distinctive 5' end pile-ups at specific sites, in spite of random distribution of reads generated from sonication, can be used to identify PT modification.

Towards this end, we devised a modification score for every position p, which is defined as the relative ratio between the number of 5' end reads at a position p and the median number of 5' end reads within a +/- 50 bp window from the position p. We complemented this modification score with a strategy for the EcoWI-seq signal deconvolution step to call PT modified positions in genomes (S1B Fig). This calculation allows for the identification of PT modification at single base resolution. In agreements with previous results [6,17,18] the profile of modification scores in the Dnd+ WT *Sen*C87 genome demonstrated that the majority of GAAC/GTTC motifs were either unmodified or modified at a very low level (modification score <5) (Figs 2B and S2A). In total, we identified 3973 predicted PT-modified sites in the WT and only one predicted modified site in the genome of the Δ*dndE* mutant strain (Figs 2C and S2B, and S1 and S2 Tables). We confirmed the reproducibility of the method in the Dnd+ WT *Sen*C87 strain with independent replicates showing a good correlation (Pearson's coefficient = 0.95, S2C Fig). Furthermore, PT-modified positions identified in the two replicates displayed 97% overlap.

Among the PT-modified sites in the genome of the Dnd+ WT *Sen*C87 strain, 97.4% (3870) were located at the GAAC/GTTC motif, indicating the precision of EcoWI in recognizing PT modified consensus sequence (Fig 2C). The PT-modified GAAC/GTTC sites account for 12% of all possible GAAC/GTTC sites (32795 in total), which is also in accordance with previous observations [16,17].

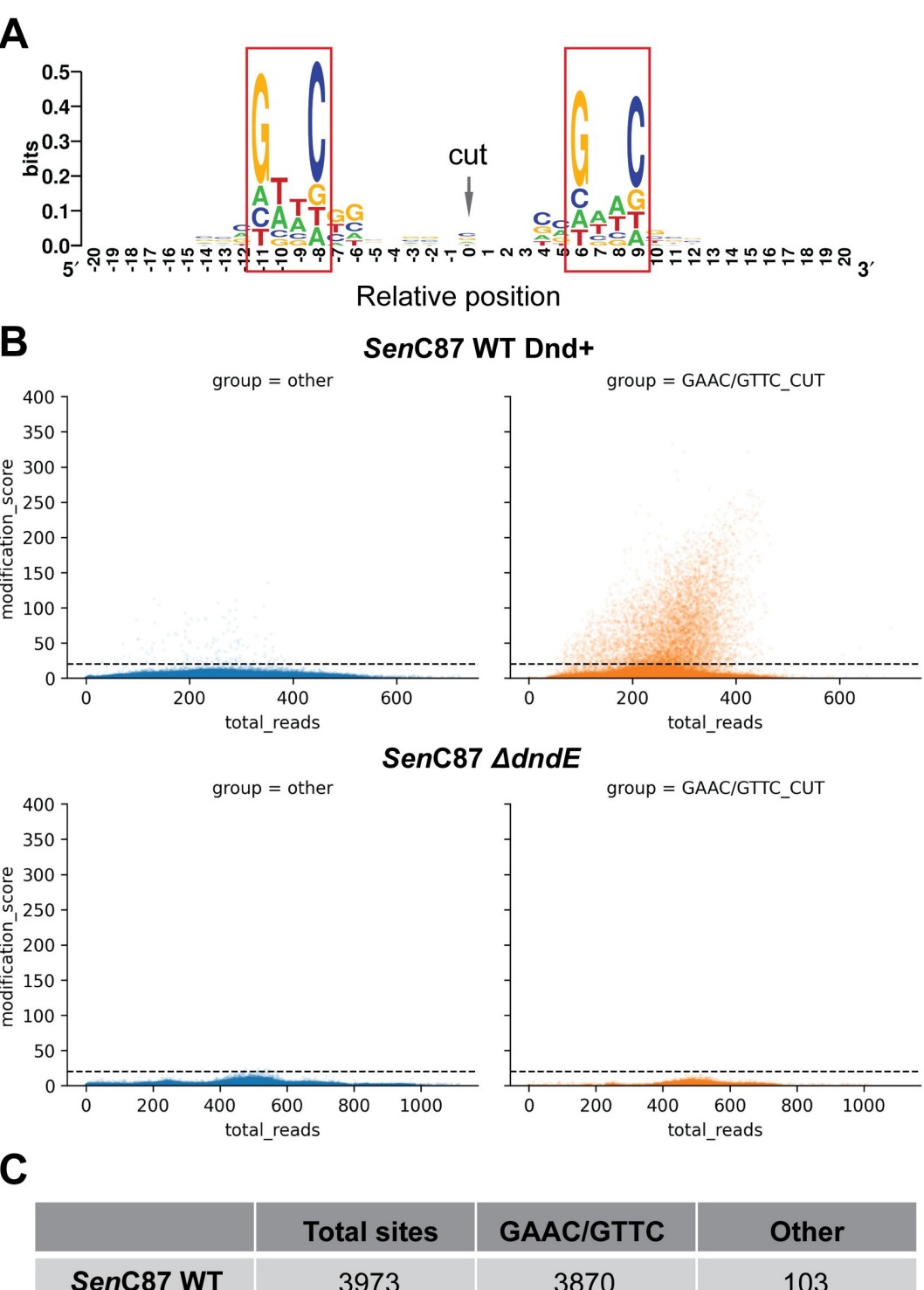

**Fig 2. Genomic mapping of PT modification at single base resolution by EcoWI-seq.** (**A**) Motif logo of positions with more than 50 5' end reads in the genome of WT Dnd+ *Sen*C87 strain. (**B**) Plots of modification scores in the genome of WT Dnd+ (top) and Δ*dndE* mutant (bottom) strains. Dashed line showed the modification score cut-off used to call PT-modified sites. (**C**) Numbers of identified PT-modified sites by category.

To understand the remaining 2.6% of modified sites located at positions other than the canonical GAAC/GTTC motif, motif logo analysis was performed on these non-canonical positions and a GTAC motif was identified (S2D Fig). Interestingly, a previous study also showed that GpsTAC accounted for 6% of *Sen*C87 PT modified sites, which was the third highest motif after the canonical GpsAAC and GpsTTC motifs [6]. It had been previously reported that some restriction enzymes (e.g. BamHI) could cleave some related sites by star activity [26]. To investigate if this is the case of EcoWI, we designed oligonucleotides containing modified GpsTAC motif or unmodified GTAC motif as control and performed EcoWI cleavage assay (S2E Fig). The GpsTAC oligonucleotide is partially cut while the unmodified GTAC oligonucleotide is not digested, demonstrating that EcoWI has activity on modified GpsTAC but not on unmodified GTAC motif at high enzyme concentration.

## 3. Validation of EcoWI-seq identified PT modification positions with SMRT sequencing

To validate the EcoWI-seq method, SMRT sequencing was performed on the same WT Dnd+ *Sen*C87 strain gDNA sample used for EcoWI-seq. We recorded the interpulse durations (IPDs) of the polymerase kinetics and followed the analytical method that has previously been published for detecting PT modification in SMRT sequencing data [16] (S3A Fig and S3 Table). As expected, we observed high kinetic signals consistently occurring at GAAC/GTTC context and obtained 3473 GAAC/GTTC sites called as modified. When comparing the specific modified positions called by EcoWI-seq and SMRT sequencing, we found 1) that the total number of modified sites identified by the two methods are very similar (3473 compared to 3973) (Fig 3); 2) a very high agreement at base resolution between the two detection methods (98.1% of SMRT modified positions are also detected in EcoWI-seq, p value = $10^{-9197.425}$, hypergeometric test) (Fig 3); and 3) that the distributions of PT modified positions identified by the two methods are found across the genome (S3B Fig). We also searched for residual EcoWI-seq signals on SMRT unmodified positions and, in agreement with the absence of modification at these positions, found no increase of read piled-up in EcoWI-seq (S3C Fig).

We confirmed that 40% (30 out of 76, p value = 4.13e-18, hypergeometric test) of the GTAC sites identified as modified in EcoWI-seq also possessed high kinetic signals in SMRT sequencing. Therefore, it is possible that this small fraction of non-canonical GAAC/GTTC "modified" sites by EcoWI-seq was attributed to the complex effect of relaxed recognition of EcoWI at preferably modified GpsTAC sequences. However, the accuracy of EcoWI-seq was not undermined as shown in the S3D Fig.

Together, these data validated the sensitivity and accuracy of EcoWI-seq in detection of PT modifications with another standard method–SMRT sequencing.

## 4. Optimization and characterization of EcoWI-seq

In efforts to optimize the EcoWI-seq performance, we constructed EcoWI-seq libraries using gDNA isolated from WT Dnd+ *Sen*C87 strain and adjusted incubation time and DNA amounts. We also analyzed data with distinct sequencing coverages and sequencing methods (single end vs paired end). As shown in S4A Fig, the number of modified sites detected by EcoWI-seq increased with prolonged enzyme treatment. The majority of modified sites can be detected with 3 hours incubation and demonstrated over 93% consistency with SMRT sequencing called modification sites (S4A and S4B Fig). Testing of library constructions suggested that EcoWI-seq was able to achieve consistent and most modification sites with a minimized 500 ng DNA input (S4C Fig). In order to get optimal performance, a minimal coverage of 200x was required in the sequencing (S4D Fig). We also investigated whether single-end vs

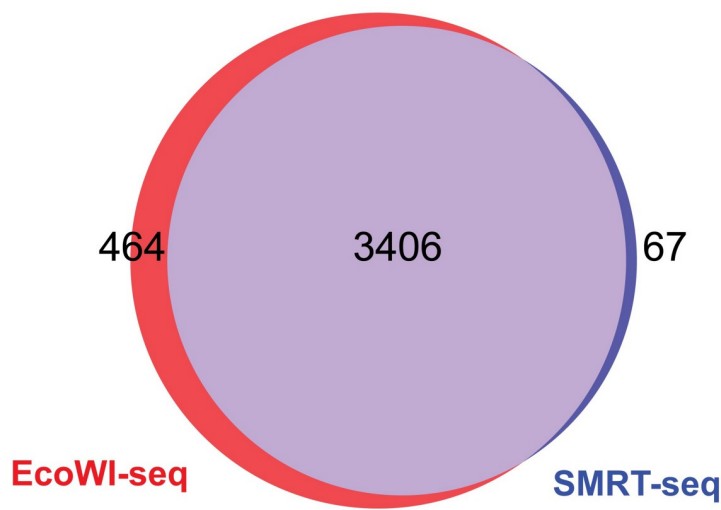

| EcoWI-seq sites | SMRT-seq sites | Overlap sites |
|:---:|:---:|:---:|
| 3870 | 3473 | 3406 (98.1%) |
| 3870 | 3473 (random) | 424 (12.2%) |

**Fig 3. Validation of EcoWI-seq called PT modification positions in the WT Dnd+ *Sen*C87 strain.** Venn plot showing the overlap of modified GpsAAC/GpsTTC positions identified using EcoWI-seq (red) and SMRT sequencing (blue). The numbers of called modified sites and overlaps were listed in the table below. The percentages of SMRT-sequencing sites overlapping with EcoWI-seq sites are listed in parentheses. Note that we also randomly chose 3473 positions from available GAAC/GTTC positions and this only led to a small percentage of overlap with EcoWI-seq modified positions.

paired-end sequencing affect the identification of PT modified sites. Our results showed that the total number of predicted modified sites and the accuracy of the predictions (validated by SMRT sequencing) are similar for single-end and paired-end sequencing. These data demonstrated the consistent performance of EcoWI-seq analysis regardless of sequencing method, therefore allowing flexibility in the selection of sequencing options (S4E Fig).

## 5. EcoWI-seq can be applied to de-novo assembled genome

To explore whether EcoWI-seq analysis can be applied to genomic DNA for which no reference genome is available, we directly assembled the EcoWI-seq sequencing reads from experiments done on WT Dnd+ *Sen*C87 strain using the *de novo* assembler SPAdes [27]. We first confirmed the quality of the assembly which demonstrated a span of nearly the size of the actual genome (S5A Fig). Next, reads were mapped back to the *de novo* assembled genome and PT modification positions were identified. Locations of modified positions found in the genome assembly show a 97.6% concordance compared to using the reference genome (S5B Fig). Therefore, EcoWI-seq can be used with similar accuracy on bacteria for which a reference genome is unavailable.

## 6. PT modification occur at consistent locations while maintaining balance

It is known from this and previous work, that only a small fraction of GAAC/GTTC motifs are PT modified. While the plasticity of PT modification in epigenomes in certain strains of

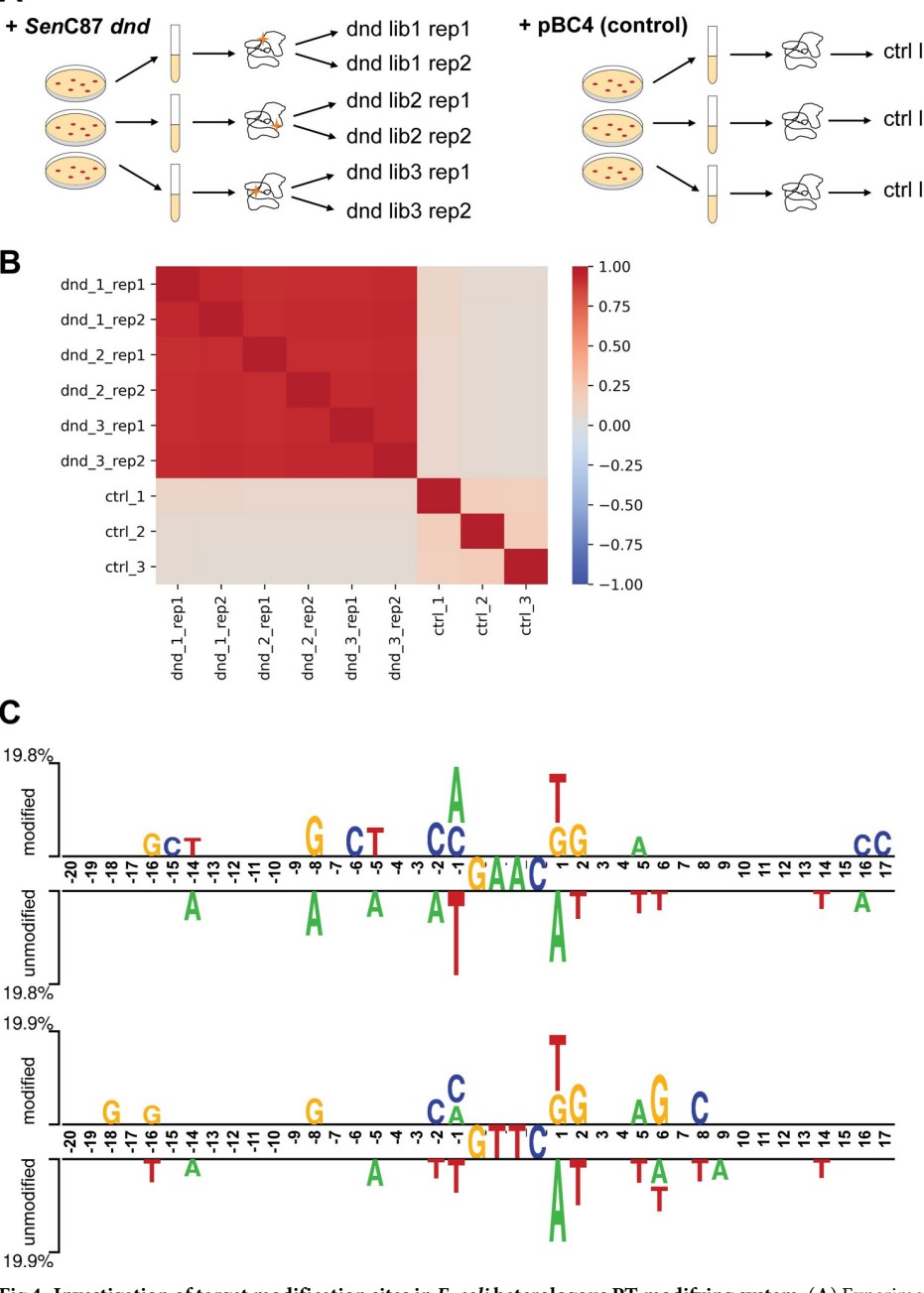

**Fig 4. Investigation of target modification sites in *E. coli* heterologous PT modifying system.** (**A**) Experiment design. Stars represent PT modifications in the genome. (**B**) Correlations among the 6 *Sen*C87 *dnd*/pBC4 transformed *E. coli* strains. For each pair of comparisons, modification scores of all possible GAAC/GTTC sites were used for correlation calculation. Color bar on the side indicated the Pearson correlation values. For *Sen*C87 *dnd* transformed *E. coli* strains, correlations were also performed among the technical duplicates (rep1 and rep2). (**C**) Output of Two Sample Logo [31]: Graphical representation showing statistically significant (p value < 0.01) differences in position-specific symbol compositions between modified and unmodified GAAC positions (top) and modified and unmodified GTTC positions (bottom) in *E. coli*.

bacteria has been investigated [6,7], it is still unclear how PT-modifying Dnd proteins select the targets and whether or not the establishment of modified positions is a stochastic process that is subsequently maintained. To address this question, we use a previously established *in*

*vivo* PT modifying system for which the expression of the *Sen*C87 *dnd* gene clusters in *E. coli* leads to a *dnd*+ strain [21]. To ensure the independent establishment of PT modifications, we conducted three independent transformation experiments respectively with *Sen*C87 *dnd* or pBC4 (pUC19 with an adenovirus DNA insert). We also included two technical replicates for each *Sen*C87 *dnd* transformed strain to rule out possible variations due to technical reasons (Fig 4A).

We found that modifications were detected on a small subset of GAAC/GTTC sites in *Sen*C87 *dnd* transformed *E. coli* gDNA (S6A and S6B Fig). This result indicates that the expression of the *dnd* gene in *E. coli* results in a genomic pattern of PT modifications similar to the *Sen*C87 strain. As expected, no modification position was identified in cultures transformed with control plasmid pBC4 (S6A and S6B Fig).

Comparison of modification scores at GAAC/GTTC positions for the three independent transformations reveals that the modified GAAC/GTTC positions are highly correlated among the three *Sen*C87 *dnd* transformed *E. coli* samples. Moreover, the correlation levels between independent transformations are equivalent to the correlation levels of the technical replicates indicating that the establishment of modified positions is non-random (Fig 4B).

If the establishment of modified positions is deterministic, we sought to identify the factor (s) that controls which positions are modified starting with possible sequence context additional to the GAAC/GTTC motifs. Towards this end, we analyzed the sequence variability (+/- 20 bp) between PT modified and unmodified GAAC/GTTC motifs in *Sen*C87 *dnd* transformed *E. coli* strains and observed nucleotide preferences immediately upstream and downstream of the modified GAAC/GTTC motif with C and G respectively are over-represented in the modified motifs at -2 and +2 positions relative to the modified compared to unmodified GAAC/GTTC motifs. The -1 bp and +1 bp positions at modified GAAC/GTTC motifs have preference for M (A or C) and K (T or G) respectively (Fig 4C). Similar sequence preferences also applied to the PT modified positions in the genome of the *Sen*C87 WT strain, suggesting a common mechanism in selection of modification sites despite the different targeting genomes (S6C Fig).

## Discussion

Phosphorothioate (PT) modification is the first DNA backbone modification identified in bacterial genomes. The ability to investigate the genomic distribution of PT modified sites is critical for revealing the potential biological functions endowed by this epigenetic system. Motivated by our previous discovery and characterization of the EcoWI enzyme which consistently and precisely cleaves at modified GpsAAC/GpsTTC motifs in synthetic oligos and plasmids containing PT modifications [21], we developed this enzymatic assay coupled with high-throughput NGS to reveal the PT genomic landscape at GAAC/GTTC sites for any bacteria. Based on the cut sites determined in the synthetic PT oligos, under partial digestion condition, EcoWI preferentially generates a double-stranded break downstream of hemi- or full-modified GTTC/GAAC motif [21]. It is possible that an additional nicking or dsDNA break occurs in the upstream of the modified motif when the enzyme concentration is increased in the assay. In the former case, this additional nicking will be repaired during illumina library preparation and leave no scar to obscure the EcoWI-induced pile-up calling. In the latter case, the consecutive occurrence of two dsDNA breaks around a modified site would lead to the loss of the ~17 bp fragment. Nonetheless, the digested pattern will still be captured by the EcoWI-seq analytical procedure using the read pile-ups in the boundary of the missing ~17 bp fragment. We rarely observed the missing ~17 bp fragment across the digested *Sen*C87 or *E. coli dnd*+ genomes, suggesting the enzyme concentration used in EcoWI-seq was within partial

digestion range. Although we expect EcoWI-seq method to detect both hemi- and full-modifications, the *SenC*87 genome used in the study has been proved to be highly biased towards fully PT-modified sites [6,18], therefore, the performance of this method on hemi-modified genomes would require further investigation.

As a novel genomic PT mapping method, EcoWI-seq have the following advantages: first, only 500 ng of gDNA is required for each assay while other published methods require at least several micrograms [16,17]. Sequencing is coupled with a convenient one-step enzymatic treatment for sample preparation and a minimal 200 x sequencing coverage requirement. These features substantially increase the feasibility of this method to be applied to various types of samples and give researchers the flexibility to determine the appropriate sequencing strategy. Second, control experiments by either omitting the EcoWI restriction step or using unmodified genomes are not necessary (as shown in S2B, S1A, and S6A Figs). Indeed, the distribution of modification scores detected in these controls were folds smaller compared to the modification scores obtained in the PT+ sample, thus leading to no or few of sites reaching the cutoff to be called as modified. Furthermore, the cutoff is sample-independent as the signal is normalized within a +/- 50 bp window to adjust for variations between different samples and sequencing depth. Third, EcoWI-seq is capable of mapping PT modifications in *de novo* assembled genomes. In such cases, even if a reference genome is not available for the sample to be examined, the genome assembly and PT modification mapping can be performed simultaneously using sequences obtained by EcoWI-seq. This allows efficient characterization of non-reference genomes and is critical for exploratory research to discover and characterize novel PT modification containing organisms. Since Human gut bacterial DNA has been reported to contain rich PT modified sites with various sequence configuration we envision the application of EcoWI-seq or PTDR-seq (PT-dependent restriction enzyme-seq) to metagenomic DNA fragments that have been enriched by binding-proficient and cleavage-deficient SprMcrA (GpsAAC, GpsTTC) to study the diversity of PT modifications in human gut microbiome [28].

One limitation of the EcoWI-seq is that this method only recognizes PT modification in the GpsAAC/GpsTTC motif due to the specificity of the EcoWI enzyme. The activity of EcoWI on modified GpsTAC suggests either a slight relaxation of PT-dependent endonuclease activity or an intrinsic property of the enzyme. So far, four types of consensus sequences for PT modification have been identified in bacteria and archaea, namely, GpsAAC/GpsTTC, GpsGCC, GpsATC and CpsCA (hemi-modified on one strand only) [3]. EcoWI-seq can only be used on bacteria with GAAC/GTTC consensus sequences for PT modification which accounts for ~37% of characterized PT containing bacteria strains (7 out of 19 strains, S4 Table) (Wang et al. 2011; Cao et al. 2014; Wu et al. 2020). The SBD of the PT-dependent endonuclease SprMcrA displays promiscuous sequence specificity, targeting full-modified GpsGCC, GpsAAC/GpsTTC, GpsATC, and hemi-modified GpsAAC/GTTC, GpsATC/GATC and GpsGCC/GGCC but not GpsTTC or CpsCA. However, the cleavage distance is variable [25,29]. To expand the usage of EcoWI-seq to multiple PT consensus motifs, it will be of great significance to find homologous endonucleases which either recognize multiple PT modified motifs (like SprMcrA) with precise cleavage patterns or show no context specificity (only recognize the P-S bond [30]). Alternative solutions are to develop enzyme mixture treatment consisting of several endonucleases with different sequence specificity or to engineer the linker region between SBD and HNH to make SprMcrA cleavage more precise. It is also possible to construct chimeric PT-dependent REase that fuses the SBD of ScoMcrA, the spacer and the HNH endonuclease domain of EcoWI, potentially cleaving GpsGCC targets with precision.

In this work, we used EcoWI-seq for base resolution mapping of PT modification in a naturally occurring PT modified genome (*SenC*87) and a heterologous synthesized PT modifying

genome (*SenC87 dnd* transferred into *E. coli* cells). Similar features of PT modifying behavior were displayed in both systems and as previously reported [3,6,16]. For example, *dnd*+ genomes are partially modified and the rate of modification is less than 15% of the available GAAC/GTTC sites. Despite the small percent of modification sites, the distribution of such modifications seems to be sporadically spread across the entire genome. However, it is interesting that biological repeats of expressing *SenC87 dnd* clusters result in a consistent set of target PT modification positions. This suggests that Dnd proteins establish the modification in a deterministic rather than a stochastic route. How does the Dnd system distinguish which positions to modify remains a mystery. Possible hypotheses are: 1) sequence-context related selection; 2) density regulation by the level of Dnd protein abundance; 3) DNA 3D structure may play a role; 4) competition between DNA methylases, such as Dam methylase which modified the GATC sites to become G6mATC [6], frequent cytosine methylase that modifies the CCD (D = A, G, T, not C) sites vs CpsCA, and Dcm methylase that modifies CCAGG sites to become Cm5CAGG vs CpsCA; 5) other mechanisms such as RNA-protein interaction guided targeting. By analyzing the differential sequences containing the modified or unmodified sites in the two PT genomes (*SenC87* and *E. coli* with transferred *dnd*), we observed similar but not identical sequence biases in proximity of the modified GAAC/GTTC sites compared to the unmodified sites (Figs 4C and S6C). Since the *dnd cluster* gene in these two systems came from the same source (*SenC87 dndBCDE*), it is possible that the difference in the sequence bias is related to the different genome composition in *Salmonella* versus *E. coli*. In addition, the favorable CG-rich regions near the modified GAAC/GTTC (especially indicated by *SenC87* modified sites) may modulate the DNA configuration for selective binding of Dnd proteins. Similar implication has been proposed in another study [6]. The additional flanking sequence preference (5' A or C [GAAC/GTTC] T or G) in the PT-modified sites suggest that the *SenC87* Dnd modification complex may select targets for modification not by 4 bp (GAAC/GTTC) specificity, but possibly by 5 bp target recognition, which partially explain the longstanding puzzle that only one modified PT site is found in approximately 1 kb DNA (this work: 3870 PT modified sites/4.6 mbp, ~0.84 PT modified sites in 1 kb).

In summary, we established a novel enzymatic method EcoWI-seq for mapping PT modification in bacterial genomes at single base resolution. EcoWI-seq is a useful technique for investigating PT modification related studies and provides a strategy for unraveling the selective mechanisms of Dnd targeting system and potential biological functions of PT modification.

## Method and materials

### Bacteria strains, plasmids and Enzymes

The wild-type *Salmonella enterica* Cerro 87 strain was kindly provided by Dr. Toshiyki Murase. The gDNA of Δ*dndE* mutant *Salmonella enterica* Cerro 87 was kindly provided by Dr. Xinyi He (mutant gDNA, Shanghai Jiaotong University). The mutant gDNA was purified by a FastPure bacteria DNA isolation kit (Vazyme #DC103). The *dnd* gene clusters responsible for *E. coli* PT modification were cloned from *Salmonella enterica* Cerro 87 into a pLacZZ (modified from pUC19) backbone and validated as described previously [21]. pLacZZ-dnd+ and the control plasmid pBC4 (pUC19 with an adenovirus DNA insert) were transformed into *E. coli* T7 Express competent cells (Dcm-, NEB #C2566) and plated on agar plates with ampicillin selection (100 μg/ml). Plasmid transformation was carried out according to the manufacturer's protocol (NEB). The purification of PT-dependent restriction enzyme EcoWI was performed as previously described [21].

### Genomic DNA extraction and plasmid isolation

For *Sen*C87 gDNA extraction, 50 mL cell culture were grown in LB to stationary phase and processed with the NEB Monarch genomic DNA purification kit (NEB #T3010). For genomic DNA extraction of *E. coli* cells, 1 ml overnight culture of pLacZZ-*dnd*+ or pBC4 transformed *E. coli* C2566 cells was harvested by centrifuging at 12,000 g for 1 min. The cell pellet was resuspended in 10 μl lysozyme (25 mg/ml) and 190 μl TE buffer, following incubation at 37˚C for 5 min. Then the lysis buffer containing 265 μl TE, 15 μl 20% SDS and 3 μl 20 mg/ml Proteinase K (NEB #P8107) was added to the sample and the mixture was incubated at 37˚C for 30 min with agitation at 1400 rpm to completely lyze the cells. An additional 2 μl RNase A (20 mg/ml, NEB #T3018) was added and followed by 5 min incubation at 56˚C to remove residual RNA in the sample. Genomic DNA was extracted with 2 rounds of equal volume phenol/chloroform extraction, one round of equal volume chloroform, precipitated in 95% ethanol and washed with 70% ethanol. The genomic DNA was dissolved in TE buffer (pH 8). For plasmid isolation, 1–3 ml of overnight culture was pelleted, and plasmid DNA was isolated using Monarch plasmid miniprep kit (NEB #T1010).

### Library preparation for EcoWI-seq

One μg genomic DNA was used for each reaction. The pLacZZ-dnd+ plasmid was spiked in as positive control. Lambda gDNA spike-in were added to serve as negative control. The genomic DNA was first digested by 1 μl EcoWI (2.7 mg/ml) in 50 μl 1x NEBuffer 2 for 3 hours of incubation at 37˚C. The digested DNA fragments were purified with oligo clean and concentrator kit (Zymo Research D4060), further sheared using 200 bp protocol with Covaris S2 Ultrasonicator in 130 μl of 0.1x TE buffer and evaporated to desired volume for illumina library preparation. One reaction of NEBNext Ultra II DNA Library Prep Kit for Illumina (NEB #E7645) was used for 1 μg genomic DNA input. The DNA library was purified with 1x volume of NEBNext Sample Purification Beads (NEB #E7103) and eluted with 40 μl of 0.1 X TE buffer.

### Illumina sequencing and reads processing

Libraries were indexed with NEBNext Multiple Oligos for Illumina (NEB #E6440) and amplified using NEBNext Ultra II Q5 Master Mix for 5 cycles. The quality and quantity of libraries were checked with Agilent TapeStation Systems with High Sensitivity D1000 reagent and screen tape (Agilent #5067–5585 and Agilent #5067–5584). The libraries were pooled for sequencing on Illumina Nextseq instrument or Illumina Novaseq instrument with paired-end reads of 75 bp or 100 bp. For each sample, we achieved 10–20 million reads.

Reads were downloaded as FASTQ files and trimmed with Trim Galore v0.6.4 (https://www.bioinformatics.babraham.ac.uk/projects/trim_galore/). The option—paired were used when processing paired end reads. Trimmed reads were aligned to the reference genome with BOWTIE2 v2.3.5.1 (26), together with SAMTOOLS V1.9 [32] to sort and index bam files for later analysis.

### Calculation of PT modification scores and calling of PT modification positions in EcoWI-seq data

To calculate the modification score per position, we first computed the read depth at each position using samtools-depth and counted the number of 5'-end reads at each position using bedtools. The read depths and 5'-end reads were split by strand. Then, the median number of 5'-end reads within a +/- 50 bp window for each position was achieved or replaced with value 1 if it resulted in 0. The modification score at position p was defined as follows: (modification

score)$_p$ = (number of 5'-end reads)$_p$ / (median number of 5' end reads)$_p$. The determination of PT modification positions in EcoWI-seq depends on two cutoffs: 1) with the reads cutoff, positions with read depths less than the reads cutoff are excluded; 2) with the modification score cutoff, we only select positions with modification scores greater than the score cutoff and their -2 bp positions must also have modification scores greater than the score cutoff. This is to avoid screening for positions generated other than double stranded breaks. Finally, minor shifts between +/- 2 bp were collapsed to the same PT modification site. In this work, we used 50 as the reads cutoff for all samples (including *Sen*C87 and *E. coli dnd*+). The choice of modification score cutoff varied from different samples: 20 was used for all *Sen*C87 WT or mutant samples and the score cutoff was set from 30–60 for *E. coli dnd*+ samples. The scripts and detailed documentation of the analysis are available at https://github.com/rosayi/EcoWI-seq.

## Library preparation for SMRT sequencing

SMRTbell libraries were constructed from *Sen*C87 gDNA samples sheared to 10–20 kb using the G-tubes protocol, end repaired, and ligated to PacBio hairpin blunt adapter. Incompletely formed SMRTbell templates and linear DNAs were digested with a combination of Exonuclease III and Exonuclease VII (NEB #M0206 and #M0379). DNA qualification and quantification were performed using the Qubit fluorometer (Invitrogen) and 2100 Bioanalyzer (Agilent Technology). The 10 kb SMRTbell library was prepared according to PacBio 8–10 kb sample preparation protocols including additional separation on a BluePippin (Sage Science), originally sequenced using C4-P6 chemistry (2 SMRT cells, 300-minute collection time).

## Detection of PT modification in SMRT sequencing data

PacBio reads were mapped to the *Sen*C87 reference genome (GenBank: CP008925.1) and mapped reads were analyzed using the SMRTProtal motif and modification analysis module version 2.3.0. IPDs were measured and processed as described [33,34] for all pulses aligned to each position in the reference sequence. The Pacific Biosciences' SMRTPortal analysis platform v. 2.3.0 uses an in silico kinetic reference, and a t-test based kinetic score detection of modified base positions (details are available at http://www.pacb.com/pdf/TN_Detecting_DNA_Base_Modifications.pdf). For calling of PT modified GAAC/GTTC sites, we plotted the kinetic signals at each GAAC/GTTC site on opposite DNA strands, and applied the same strategy as previously described [16] (see S3A Fig).

## Comparison of modified positions

Candidate PT-modified GAAC/GTTC positions called from EcoWI-seq and SMRT sequencing methods were mapped to *Sen*C87 reference genome (GenBank: CP008925.1) using bedtools. Overlaps between two datasets were calculated with bedtools command line or custom python script. For comparison of *E. coli* heterologous PT modifying samples, Pearson's correlation was performed for each pair of samples on all GAAC/GTTC sites (including both modified and unmodified) and the correlation heatmap was plotted with python seaborn module.

## Motif logo analysis

The sequences were extracted between 20 bp upstream and 20 bp downstream of a modification site and nucleotide preferences were visualized using the weblogo web tool (https://weblogo.berkeley.edu/logo.cgi) [35]. For the visualization of statistically significant nucleotides bias around modified and unmodified GAAC/GTTC sites, we generated two datasets by extracting and aligning sequences between +/- 20 bp of modified GpsAAC/GpsTTC and

unmodified GAAC/GTTC sites respectively. Two sample logos (http://thetruth.ccs.neu.edu/cgi-bin/tsl/tsl.cgi) [36] was used to determine and visualize the significant differences in nucleotide composition under a t-test with p value set to 0.01.

### *De novo* assembly

Genome *de novo* assembly was performed with SPAdes v3.13.0 [27] with default options. The correspondence between modified positions identified in the *de novo* assembled genome and the reference *Sen*C87 genomes was done using the liftover tool flo [37].

### EcoWI digestion of PT-modified oligos (GpsTAC/GpsTAC)

58mer deoxyoligonucleotides (oligos) were purchased from IDT (Integrated DNA Technologies. USA). Below is shown the DNA sequences:

Top: 5'-CCTCGAGGTCGACGGTATCGCCCGATGpsTACCTAGTCGAATTCCTGCAGCCCGGGGGA-3'

Bottom: 5'-TCCCCCGGGCTGCAGGAATTCGACTAGGpsTACATCGGGCGATACCGTCGACCTCGAGG-3'

10 ng of the duplex DNA oligos (9 nM) containing PT-modified site (GpsTAC/GpsTAC) or unmodified site (GTAC/GTAC) were digested by 5.1 μg (76 μM) and 1.7 μg (26 μM) of EcoWI in NEB Buffer 2.1 for 2 h at 37˚C in a 30 μl reaction volume. The digest mix was treated with 4 μl of Proteinase K (3.2 U, NEB) at room temperature for 20 min. Half of the digested DNA was then analyzed by PAGE on a 10% TBE-PAG (Thermo-Fisher). DNA fragments were stained by SYBR Gold (Thermo-Fisher) and imaged on a Typhoon imager (GE Health). The same duplex oligos were also digested by EcoRI-HF (RI, 40 U, recognition sequence GAATTC, NEB) as a positive digestion control for 2 h at 37˚C in NEB CutSmart buffer. 5-bp DNA ladder was purchased from Thermo-Fisher.

## Supporting information

**S1 Fig. Development of EcoWI-seq.** (**A**) Plots of the 5' end reads coverage across all genomic positions. Top: EcoWI-seq on WT Dnd+ *Sen* 87 strain; middle: EcoWI-seq on Δ*dndE* mutant strain; bottom: Sequencing of WT Dnd+ *Sen*C87 strain without EcoWI treatment. (**B**) Schema describing the analytical steps for PT-modified sites identification using EcoWI-seq. (TIF)

**S2 Fig. Genomic mapping of PT modification at single base resolution by EcoWI-seq.** (**A**) Frequencies of modification level at all possible GAAC/GTTC sites for WT Dnd+ (orange) and Δ*dndE* mutant (blue) *Sen*C87 strains. (**B**) Curve showing the number of PT modified sites identified at different modification score cutoff. The cutoff for calling modified sites in this experiment was determined by visual observation of the deviation from the tangent point. (value = 20). (**C**) Scatter plot showing correlation of modification scores in two independent replicates. Each dot represents a GAAC/GTTC site. (**D**) Motif logo of the flanking regions of non-canonical GAAC/GTTC sites. Positions are relative to the predicted cutting site (position 0). (E) EcoWI digestion of PT-modified duplex oligos containing GpsTAC or unmodified GTAC site. S, P1, and P2 refer to substrate (58mer), cleavage products 1 and 2, respectively. EcoWI digest condition, PAGE analysis, DNA fragment staining and imaging were described in the Materials and Method. EcoRI-HF (RI) was used as a positive control for restriction. (TIF)

**S3 Fig. Validation of PT modification positions identified using EcoWI-seq.** (**A**) Kinetic signals of GAAC/GTTC sites across the genome of the WT Dnd+ *SenC87* strain using SMRT-

sequencing. The dashed line displayed the threshold cutoff for calling modified sites (16). (B) Genome-wide circos plot showing distribution of PT modifications in WT Dnd+ SenC87 strain. The height of the bars represented the modification scores in EcoWI-seq (red) and kinetics signals in SMRT sequencing (blue). Only called modification sites were displayed. (C) Plots showing number of 5' end reads in EcoWI-seq data across GAAC/GTTC motifs. Positions were relative to the GAAC/GTTC motif (+0 was set as the position of the G). Top: EcoWI-seq signal in the WT Dnd+ strain for the 3473 GAAC/GTTC sites called as modified by SMRT sequencing; middle: EcoWI-seq signal in the WT Dnd+ strain for 3500 randomly selected sites called as unmodified by SMRT sequencing; bottom: EcoWI-seq signal in the *ΔdndE* mutant strain for the 3473 GAAC/GTTC sites called as modified by SMRT sequencing using the WT Dnd+ strain. (D) Curve with Y axis represents the overlapping percent of modified sites called by EcoWI-seq and SMRT sequencing (percentage relative to number of modified sites in SMRT sequencing) and X axis represents the percent of identified non-canonical GAAC/GTTC modification sites in EcoWI-seq. Data points indicated the threshold cutoff used.
(TIF)

**S4 Fig. Optimization and characterization of EcoWI-seq.** (**A**) Plot showing increased numbers of modified sites with increased incubation time. Data points represented 1, 2, 3 and 6 hours enzymatic treatments respectively. (**B**) Numbers of modified GAAC/GTTC sites and overlap percentage with different incubation conditions. Percent SMRT: percentage of SMRT-seq sites overlapping with EcoWI-seq sites; Percent EcoWI: Percentage of EcoWI sites overlapping with SMRT-seq sites. (**C**) Plot showing numbers of modified sites over different amounts of DNA input. Data points indicated DNA input of 250 ng, 500 ng and 1000 ng respectively. (**D**) Plot showing numbers of modified sites using different sequencing depth. Data points represented an average coverage of 100x, 150x, 200x, 234x, 300x and 335x respectively. (**E**) Numbers of modified GAAC/GTTC sites and overlap percentages by single end or paired-end sequencing. Reads were down-sampled to the same numbers for comparison in the two conditions. Overlap percent: Percentage of EcoWI sites overlapping with SMRT-seq sites.
(TIF)

**S5 Fig. Application of EcoWI-seq analysis with *de novo* assembly.** (**A**) Features of *de novo* assembly using SPAdes assembler. (**B**) Table listing numbers and overlapping of called modified sites between *de novo* assembly and using reference genome.
(TIF)

**S6 Fig. Establishment of PT modification system in heterologous host *E. coli*.** (**A**) Plots of modification scores in the genome of *Sen*C87 *dnd* transformed and pBC4 transformed *E. coli* strains. Dashed line showed the modification score cutoff used to call PT modified sites. (**B**) Numbers of identified PT-modified sites by category. (**C**) Output of Two Sample Logo: Graphical representation showing statistically significant (p value < 0.01) differences in position-specific symbol compositions between modified and unmodified GAAC positions (top) and modified and unmodified GTTC positions (bottom) in the *Sen*C87 genome.
(TIF)

**S1 Table. Modification scores of every base in WT *Sen*C87 by EcoWI-seq analysis.**
(ZIP)

**S2 Table. PT modified positions called by EcoWI-seq analysis in WT *Sen*C87.** This file can be open using any free text editors.
(SITES)

**S3 Table. Kinetic signals from SMRT sequencing at modified bases.**
(GFF)

**S4 Table. Bacteria strains containing PT modification and their corresponding PT-modified consensus motifs.** This is a.CSV file that can be open in any free text editors or Microsoft office.
(CSV)

## Acknowledgments

We thank Dr. Toshiyki Murase for sending the wild type *Sen*C87 strain and Dr. Xinyi He and his student at Shanghai Jiaotong University for providing the Δ*dndE Sen*C87 gDNA. We are grateful to Dr. Vladimir Potapov for his assistance in PacBio data analysis and Dr. Peter Dedon of MIT for the initial characterization of PT-modified plasmids.

## Author Contributions

**Conceptualization:** Weiwei Yang, Shuang-yong Xu, Laurence Ettwiller.

**Data curation:** Weiwei Yang.

**Formal analysis:** Weiwei Yang.

**Investigation:** Weiwei Yang, Alexey Fomenkov, Dan Heiter, Shuang-yong Xu, Laurence Ettwiller.

**Methodology:** Weiwei Yang, Dan Heiter, Shuang-yong Xu, Laurence Ettwiller.

**Project administration:** Laurence Ettwiller.

**Resources:** Alexey Fomenkov, Dan Heiter, Shuang-yong Xu, Laurence Ettwiller.

**Software:** Weiwei Yang.

**Supervision:** Shuang-yong Xu, Laurence Ettwiller.

**Validation:** Weiwei Yang, Dan Heiter.

**Visualization:** Weiwei Yang.

**Writing – original draft:** Weiwei Yang, Laurence Ettwiller.

**Writing – review & editing:** Weiwei Yang, Shuang-yong Xu, Laurence Ettwiller.

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
