## [Decision Letter · Decision Letter 0]

19 Jul 2022

Dear Dr Ettwiller,

Thank you very much for submitting your Methods entitled 'High-throughput sequencing of EcoWI restriction fragments maps the genome-wide landscape of phosphorothioate modification at base resolution.' to PLOS Genetics.

The manuscript was fully evaluated at the editorial level and by independent peer reviewers. The reviewers appreciated the attention to an important topic but identified some concerns that we ask you address in a revised manuscript

We therefore ask you to modify the manuscript according to the review recommendations. Your revisions should address the specific points made by each reviewer.

[LINK]

Yours sincerely,

Chunxiao Song

Guest Editor

PLOS Genetics

Quanjiang Ji

Section Editor: Methods

PLOS Genetics

Reviewer's Responses to Questions

**Comments to the Authors:**

Reviewer #1: In this manuscript, the author developed EcoWI-seq based on a modification-dependent restriction endonuclease EcoWI to profile the phosphorothioation modifications in the genomic DNA. In the previous work, they characterized EcoWI as a PTDR to cleave the DNA strand at a fixed distance from its recognition sequence GpsAAC/GpsTTC. Here, they extended a straightforward application of EcoWI for genome-wide PT modifications identification at single base resolution combined with next generation sequencing. Similar to other sequencing methods based on various restriction endonuclease, EcoWI-seq is easy-operated and can be optimized to little starting material. The shortcoming of these kinds of method are obvious by the strand specificity of the restriction endonuclease. The comments and suggestions are listed below:

1. In the previous work, the author studied the performance of EcoWI in synthesized oligonucleotides and plasmids. It is suggested that these artificial DNAs with known modification ratios should be added to the experimental samples as spike-in. It will provide very clear evidence for the accuracy of EcoWI-seq.

2. There are mainly four types of consensus sequences for PT modification. EcoWI-seq can only detect one of them. Thus, what is the proportion of GpsAAC/GpsTTC in all sequences with PT modifications? It is better to tell the readers the percent of the detected sites among all PT modifications.

3. It is interesting that GpsTAC motif can be detected by EcoWI-seq besides the canonical GpsAAC/GpsTTC motif. The authors think that it may be attributed to the star activity of the EcoWI enzyme for either unmodified GTAC or PT modified GpsTAC motif. The biochemistry experiments should be performed to verify this possibility.

Reviewer #2: Yang et al present a manuscript titled “High-throughput sequencing of EcoWI restriction fragments maps the genome-wide landscape of phosphorothioate modification at base resolution”. The manuscript described a new technique EcoWI-seq to detect phosphorothioate (PT) modification. The method relies on the specific restriction site of EcoWI enzyme that will cut GAAC/GTTC motifs of non-PT modified sites of genomic DNA. The manuscript is very well written and demonstrate a good experimental design to show the robustness of the technique. The authors used Salmonella enterica cerro 87 as the show case and showed a high consistency result with previously published work of the same species. Moreover, the author also demonstrated the PT detection by the EcoWI-seq in the E. coli expressing the dnd+ gene cluster. The authors also provided a bioinformatic pipeline to analyze the data generated from EcoWI-seq in GitHub that will be useful for the community and for reproducibility of their work.

Comment.

- Sonication of EcoWI digested DNA could produce a background noise due to a probability of the DNA breaking at the GAAC/GTTC with PT modified sites. Please discuss the impact of sonication.

- Figure 3b is conclude nothing to me. We cannot see any details of base per base comparison due to too much information. I recommend remove from the panel main figure.

- The results from E. coli expressing the dnd+ gene cluster can be used to understand a mechanism of PT modification. I would like the authors explore more what’s happened in E coli compare to Salmonella and report in the manuscript such as i) sequence context around of PT modification sites, ii) gene that has PT modification site(s).

---

## [Editor Report · Decision Letter 1]

18 Aug 2022

Dear Dr Ettwiller,

We are pleased to inform you that your manuscript entitled "High-throughput sequencing of EcoWI restriction fragments maps the genome-wide landscape of phosphorothioate modification at base resolution." has been editorially accepted for publication in PLOS Genetics. Congratulations!

Yours sincerely,

Chunxiao Song

Guest Editor

PLOS Genetics

Quanjiang Ji

Section Editor

PLOS Genetics

Comments from the reviewers (if applicable):

**Data Deposition**

http://datadryad.org/submit?journalID=pgenetics&manu=PGENETICS-D-22-00742R1

**Press Queries**

---

## [Editor Report · Acceptance letter]

13 Sep 2022

PGENETICS-D-22-00742R1 

High-throughput sequencing of EcoWI restriction fragments maps the genome-wide landscape of phosphorothioate modification at base resolution. 

Dear Dr Ettwiller, 

We are pleased to inform you that your manuscript entitled "High-throughput sequencing of EcoWI restriction fragments maps the genome-wide landscape of phosphorothioate modification at base resolution." has been formally accepted for publication in PLOS Genetics! Your manuscript is now with our production department and you will be notified of the publication date in due course.

With kind regards,

Livia Horvath

PLOS Genetics

On behalf of:
